# FMRI-PTE: A LARGE-SCALE FMRI PRETRAINED TRANSFORMER ENCODER FOR MULTI-SUBJECT BRAIN ACTIVITY DECODING

## ABSTRACT

The exploration of brain activity and its decoding from fMRI data has been a longstanding pursuit, driven by its potential applications in brain-computer interfaces, medical diagnostics, and virtual reality. Previous approaches have primarily focused on individual subject analysis, highlighting the need for a more universal and adaptable framework, which is the core motivation behind our work. In this work, we propose fMRI-PTE, an innovative auto-encoder approach for fMRI pre-training, with a focus on addressing the challenges of varying fMRI data dimensions due to individual brain differences. Our approach involves transforming fMRI signals into unified 2D representations, ensuring consistency in dimensions and preserving distinct brain activity patterns. We introduce a novel learning strategy tailored for pre-training 2D fMRI images, enhancing the quality of reconstruction. fMRI-PTE's adaptability with image generators enables the generation of well-represented fMRI features, facilitating various downstream tasks, including within-subject and cross-subject brain activity decoding. Our contributions encompass introducing fMRI-PTE, innovative data transformation, efficient training, a novel learning strategy, and the universal applicability of our approach. Extensive experiments validate and support our claims, offering a promising foundation for further research in this domain.

## 1 INTRODUCTION

Deciphering the intricacies of brain activity involves the extraction of meaningful semantics from intricate patterns of neural activity within the brain, as highlighted by (Parthasarathy et al., 2017; Horikawa & Kamitani, 2017; Beliy et al., 2019; Shen et al., 2019). When humans are exposed to visual stimuli, the neural responses in the brain are typically quantified by monitoring changes in blood oxygenation using functional Magnetic Resonance Imaging (fMRI) (Kwong et al., 1992). This challenging task aims to unveil and comprehend the neural representations underlying diverse mental processes, sensory perceptions, and cognitive functions, all of which are inferred from the intricate dance of neurons and brain regions. Its significance is underscored by its far-reaching applications in fields such as brain-computer interfaces, medical diagnosis and treatment, and virtual reality, attracting substantial attention from both scholars and the public alike.

Remarkable endeavors have been undertaken to unravel the intricacies of brain activity through the decryption of fMRI data. In pursuit of this goal, a prevalent preprocessing strategy involves the transformation of signals originating from the visual cortex into coherent vectors. Subsequently, two distinct avenues emerge for the interpretation of fMRI data. One path involves the intricate modeling of the alignment between feature vectors originating from diverse modalities, encompassing endeavors such as mapping fMRI signals into images (Liu et al., 2023; Gu et al., 2022) or transmuting fMRI data into textual representations (Du et al., 2023; Takagi & Nishimoto, 2023). The other avenue revolves around the construction of robust generative models, their development meticulously conditioned upon fMRI signals (Chen et al., 2022; Mozafari et al., 2020). Nonetheless, an imposing challenge looms on the horizon due to the paucity of large-scale fMRI-image pairs, which impedes the establishment of a robust correlation between fMRI signals and their corresponding visual stimuli. Consequently, the reconstructed images stemming from neural activity tend to exhibit a regrettable blend of blurriness and semantic misalignment.

Furthermore, it is essential to underscore that prior fMRI decoding methodologies have been primarily tailored to individual subjects, rather than addressing the multi-subject aspect, as is the focus of this paper. This distinction is paramount, as inherent variability amongst individuals, molded by genetic predispositions, environmental influences, and life experiences, engenders discernible disparities and nuances in brain architecture (Seghier & Price, 2018). Consequently, fMRI signals manifest variations in dimensions and responses when juxtaposed across different subjects, even when they are exposed to identical visual stimuli. This inherent variance renders models trained on a single subject ill-suited for broader application to others. Notwithstanding some notable strides in enhancing the faithfulness and semantic coherence of image generation from visual stimuli (Qian et al., 2023; Chen et al., 2022), these approaches necessitate the arduous task of individually tailoring models for each subject, thereby substantially curtailing their universality and generalizability.

Therefore, our core idea centers on introducing a large-scale, pre-trained transformer encoder specifically designed for fMRI data, which we refer to as fMRI-PTE. This innovation serves as a versatile tool capable of understanding brain activity across various subjects. We draw inspiration from the impressive accomplishments of well-known pre-trained models in deep learning, including BERT (Devlin et al., 2018), MAE (He et al., 2022), and GPT (Radford et al., 2018), which have excelled in comprehending and applying knowledge across a wide range of tasks in computer vision and natural language processing.

The fundamental principle behind our fMRI-PTE closely aligns with these renowned models, primarily relying on self-supervised learning to distill valuable insights from extensive datasets. These insights can be seamlessly applied to various tasks or fine-tuned to suit specific goals. However, it is essential to note that while pre-trained models have been highly successful in effective representation learning, their application to decoding brain activity from fMRI data remains relatively unexplored, with only a few attempts in this direction. A noteworthy example is the innovative strategy introduced by (Chen et al., 2022), which involves masked brain modeling aimed at obtaining a robust self-supervised representation of fMRI signals. It is important to note that these pre-trained models often require additional fine-tuning on an individual basis to align with the intricate biological nuances governing the generation of visual stimuli. In this paper, our primary goal is to develop a more universally applicable foundational model tailored specifically for fMRI signals. This endeavor strives to establish a clear and widely applicable connection between brain activities and visual stimuli, transcending individual subject boundaries.

Based on the insights above, we introduce fMRI-PTE, a straightforward auto-encoder approach tailored for pre-training with a vast dataset of over 40,000 fMRI subjects. Its primary goal is to address the challenge arising from variations in fMRI data dimensions due to individual brain differences. To accomplish this, we propose an innovative method to transform fMRI signals into unified representations. Initially, we convert individual native-space fMRI data into a common *anatomically aligned group surface space*. Additionally, we project cortical activation flatmaps onto 2D brain activation images, as illustrated in Figure 1. This novel fMRI representation not only maintains consistent dimensions across individuals but also preserves the distinct patterns of brain activity among them. This 2D representation facilitates the efficient training of foundation models, building upon the advancements in deep learning communities. In contrast to representing fMRI data as 1D vectors, these brain activity surface maps convey richer visual semantics and explicitly capture spatial information about neural signals within the brain's structure.

Furthermore, we introduce a novel learning strategy for pre-training 2D fMRI images, which possess distinct characteristics compared to natural images. Notably, we observe that differences in brain surface images manifest in the activity texture, encompassing both low and high-frequency signals. Unlike natural images, they exhibit less spatial redundancy, which can lead to blurring in local regions when employing masked autoencoding, as depicted in Figure 3.

Specifically, our fMRI-PTE utilizes self-supervised learning through autoencoders to reconstruct input fMRI signals. Consequently, fMRI-PTE is designed to encode the original fMRI signals by compressing high-dimensional data into a low-dimensional space and then reconstructing them to their original form. We introduce a two-stage learning approach, focusing on reconstruction and compression, to enhance the quality of the reconstruction process.

Equipped with adaptable image generators, fMRI-PTE empowers us to generate well-represented fMRI features, facilitating brain activity decoding through feature alignment or conditional genera-

tion paradigms. Importantly, we can extend our capabilities to cross-subject brain activity decoding, offering an improved benchmark with superior generalization performance. Additionally, this enables the analysis of similarities and differences in brain function across individuals. Thus, we envision fMRI-PTE serving as an effective foundational framework supporting subsequent downstream tasks and inspiring further research in this domain.

We summarize our contributions outlined as follows: 1) **Introduction of fMRI-PTE**: We propose fMRI-PTE, an auto-encoder approach designed for fMRI pre-training, addressing the challenge of dimension variations in fMRI data due to individual brain differences. 2) **Innovative Data Transformation**: We introduce an innovative approach to transform fMRI signals into unified representations by projecting them onto 2D brain activation images, maintaining consistency in dimensions across individuals while preserving distinct brain activity patterns. 3) **Efficient Training**: The 2D representation facilitates efficient training of foundation models, leveraging prior advancements in deep learning communities, making it suitable for both within-subject and cross-subject brain activity decoding. 4) **Novel Learning Strategy**: We propose a novel learning strategy for pre-training 2D fMRI images, acknowledging their distinct characteristics compared to natural images. This strategy enhances the quality of reconstruction. 5) **Universal Applicability**: fMRI-PTE's adaptability with image generators empowers the generation of well-represented fMRI features, facilitating various downstream tasks, and cross-subject brain activity decoding. Extensive experiments validate and support our claims.

## 2 RELATED WORK

**Foundation models for neuroscience.** Foundation models have garnered considerable attention for their exceptional generalization abilities, as evidenced by their success in various domains (Bommasani et al., 2021; Devlin et al., 2018; Brown et al., 2020; Yuan et al., 2021; Yu et al., 2022). However, their application in neuroscience remains relatively uncharted territory. Recent efforts by Chen et al. (2022; 2023) have involved pre-training masked autoencoders on data from over 1000 subjects, followed by fine-tuning for visual decoding tasks. Nevertheless, the limited sample size constrains their potential for broader generalization. Furthermore, their pre-training focus solely on visual Regions Of Interest (ROIs), limiting their versatility for other applications. In contrast, Caro et al. (2023) harnessed masked autoencoders with large-scale fMRI data from the UK Biobank (UKB) and Human Connectome Project (HCP), showcasing their utility in clinical variable prediction and future brain state prediction. However, the use of parcel data in their pre-training phase lacks the granularity required for fine-grained tasks. A similar challenge of information loss is encountered by Thomas et al. (2022), and Ye & Pandarinath (2021), who, despite adopting a setting akin to BERT for training transformers on neural population data from monkeys, have not demonstrated significant generalization to human fMRI data as our work.

**Inter-subject neural decoding.** The exploration of inter-subject information decoding has been a subject of extensive investigation across diverse data modalities and decoding tasks (Roy et al., 2020; Wei et al., 2021; Zubarev et al., 2019; Halme & Parkkonen, 2018; Richiardi et al., 2011; Wang et al., 2020). However, the persisting limitations stemming from the scarcity of training data and considerable subject variability leave room for improvements in inter-subject decoding performance. Raz et al. (2017) have made efforts to establish robust decoding models using multiple runs of video fMRI data for coarse audio and visual feature decoding. Nonetheless, their algorithms impose constraints on their ability to effectively model subject and stimulus variability. Recent endeavors have turned to deep learning techniques in pursuit of enhanced inter-subject decoding, but they continue to grapple with the challenge of limited training data (Roy et al., 2020; Wei et al., 2021). In a different vein, Bazeille et al. (2021) have proposed the use of functional alignment to bolster inter-subject decoding. Nevertheless, the practical challenges associated with data sharing for alignment persist in many scenarios.

## 3 FMRI-PTE

Having thoroughly examined the motivation and challenges outlined above, we now formally introduce fMRI-PTE, a straightforward auto-encoder approach tailored for fMRI pre-training. This method involves encoding the initial fMRI signals into a condensed lower-dimensional space and

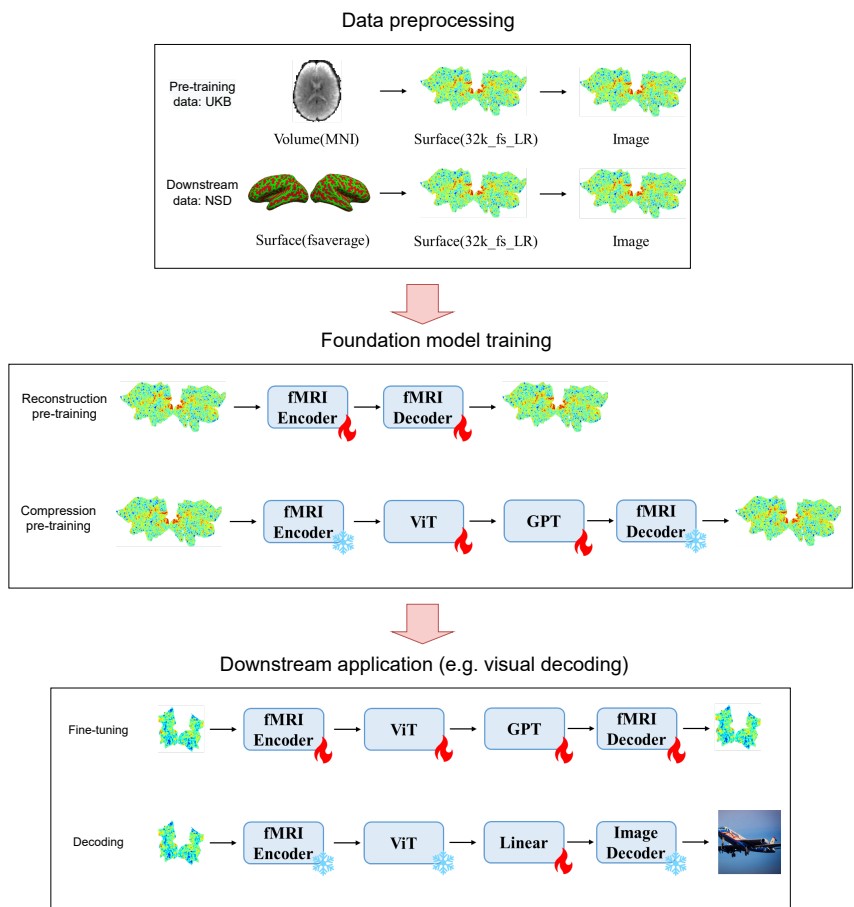

Figure 1: Pipeline for training and application of fMRI-PTE model.

subsequently decoding them to reconstruct the original signals. Serving as a foundational model, its primary purpose is to acquire cohesive representations applicable to diverse individuals, facilitating the use of compressed features in cross-subject downstream tasks. The comprehensive pipeline of our proposed fMRI foundation model is self-explanatory and meticulously depicted in Figure 1, while additional intricacies of our design are elaborated upon below.

### 3.1 UNIFIED REPRESENTATIONS FOR FMRI SIGNALS

Our pre-training dataset draws from resting-state fMRI data sourced from UK Biobank (Miller et al., 2016), and to ensure data uniformity, we harnessed pre-processed image data generated through an image-processing pipeline developed and executed on behalf of UK Biobank (Alfaro-Almagro et al., 2018). This extensive dataset comprises approximately 39,630 subjects for training, with an additional 1,000 subjects held in reserve for validation, each contributing a single session comprising 490 volumes. In a contribution to the research community, we will make the processed codes and dataset publicly available, fostering future research in this burgeoning field.

Diverging from conventional approaches that flatten fMRI voxels into 1D signals, we adopted a novel strategy that preserves the spatial relationships of voxels within each hemisphere. Leveraging surface fMRI data, we transformed this information into 2D images. For the resting-state fMRI data from UK Biobank, our process began with the conversion of

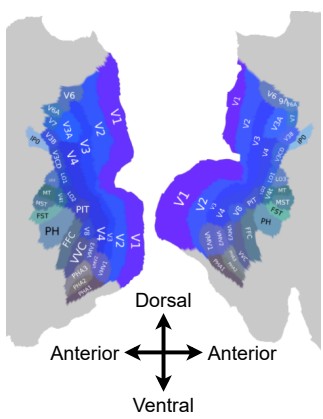

Figure 2: Visual cortical ROIs.

time series data from MNI volume space to 32k_fs_LR surface space (Glasser et al., 2013). Subsequently, for each frame within the time series, we applied z-scoring to the values across every vertex within the session. These z-scored values were then meticulously mapped onto 2D images with dimensions of $1023 \times 2514$ through the use of pycortex (Gao et al., 2015).

In the case of NSD, our methodology involved an initial conversion of GLM results from the "betas_fithrf_GLMdenoise_RR" version in fsaverage space to 32k_fs_LR space. Subsequent within-session z-scoring was applied before rendering the data into $1023 \times 2514$ 2D images.

Our approach also involved the selection of early and higher visual cortical ROIs, comprising a total of 8,405 vertices, as per the HCP-MMP atlas (Glasser et al., 2016) in the 32k_fs_LR space. These ROIs encompassed regions such as "V1, V2, V3, V3A, V3B, V3CD, V4, LO1, LO2, LO3, PIT, V4t, V6, V6A, V7, V8, PH, FFC, IP0, MT, MST, FST, VVC, VMV1, VMV2, VMV3, PHA1, PHA2, PHA3", as illustrated in Figure 2.

Furthermore, in the NSD, we optimized the training dataset by averaging repetitive fMRI trials involving the same images. This approach yielded a training dataset that featured distinct visual stimuli tailored to individual subjects, while the testing dataset was characterized by stimuli shared across all subjects, thereby facilitating robust cross-subject analyses and research.

## 3.2 AUTOENCODERS FOR fMRI COMPRESSION

The primary objective of the autoencoders is to compress the inherently high-dimensional fMRI signals into a more manageable, low-dimensional feature space, facilitating their utilization in downstream tasks across diverse individuals. However, traditional autoencoders struggle with two significant challenges: 1) *Preserving High-Frequency Signals*: The application of a high compression ratio often results in the loss of critical high-frequency signals during the reconstruction process. These high-frequency signals play a pivotal role in revealing the intricate patterns, regions, and directional aspects of signal activation. 2) *Spatial Interaction Across Brain Regions*: The signals originating from distinct regions of the brain surface can exhibit complex interactions and collaborations across spatial dimensions, adding another layer of complexity to the reconstruction process. To address these challenges, we introduce an innovative two-stage learning approach encompassing reconstruction and compression.

### 3.2.1 QUANTIZED RECONSTRUCTION STAGE

In the initial reconstruction stage, we employ a symmetrical architecture comprising an encoder ($\mathcal{E}$) and a decoder ($\mathcal{D}$). This encoder's structure draws inspiration from VQGAN (Esser et al., 2021) and consists of five blocks that iteratively reduce the resolution of the input surface image. Each block incorporates stacked residual layers to extract deep features. Notably, in this first stage, we predominantly leverage convolution layers to harness their ability to capture local patterns, including both high- and low-frequency signals within localized regions. Importantly, the features are passed from the encoder to the decoder with minimal loss, enabling us to focus primarily on the quality of reconstruction during the first stage. This approach significantly enhances our capacity to accurately reconstruct high-frequency signals.

To further optimize the extracted features from $\mathcal{E}$, we apply vector quantization (Esser et al., 2021). This prepares the features for integration with transformers in the second stage, which we will introduce shortly. The objective function encompasses a combination of perceptual loss (Zhang et al., 2018), adversarial loss (Goodfellow et al., 2020), reconstruction loss, and commitment loss (Van Den Oord et al., 2017). This multifaceted loss framework ensures the fidelity of the reconstructed signals while preserving essential details, making our approach highly effective in capturing the nuances of brain activity.

### 3.2.2 TRANSFORMER COMPRESSION

The second stage of our approach employs an autoencoder structure, strategically positioned within the first stage. As depicted in Fig. 1, this stage takes quantized feature indices as inputs and focuses on reconstructing the original indices following compression. The quantization step relieves us of the burden of reconstructing image textures, instead allowing us to delve into learning the intricate correlations among different indices. This is where the strength of transformer-like architectures

shines, as they transcend the confines of local receptive fields and excel at modeling long-range dependencies.

In this context, the encoder $\mathcal{E}_2$ adopts a ViT (Dosovitskiy et al., 2021) architecture but operates on index embeddings, each supplemented with positional information to contextualize its location within the surface image.

The output features from encoder $\mathcal{E}_2$, denoted as $f \in \mathbb{R}^{L0 \times C_0}$, undergo further compression via a mapper layer $\mathcal{M}$, transforming them into $\tilde{f} \in \mathbb{R}^{L_1 \times C_1}$. Here, $L$ and $C$ represent batch size, the number of tokens, and feature dimensions. The mapper layer $\mathcal{M}$ can be realized either as a compact network (e.g., reducing the number of tokens before feature dimension reduction) or as a slice operation (e.g., retaining only the [CLS] token embedding).

Moving to the decoder $\mathcal{D}_2$, it consists of stacked transformer blocks, resembling the MAE decoder (He et al., 2022), albeit with a 100% mask ratio. To initialize, we introduce a sequence of [MASK] tokens, represented as $[tm, t_m, \cdots, t_m] \in \mathbb{R}^{L_0 \times C_0}$, followed by the inclusion of the compressed feature $\tilde{f}$ as a conditioning element. We seamlessly align the feature dimensions between $\tilde{f}$ and $t_m$ without the need for an additional fully-connected layer.

The transformer architecture incorporates bi-directional attention, enabling interactions both within tokens and between conditions and tokens. Towards the end of the decoder, we discard the conditioning component, leaving behind the remaining tokens for the prediction of their corresponding target indices. Training the second stage entails computing the cross-entropy loss by comparing predicted probabilities against ground-truth indices, ensuring that our model effectively captures the intricate relationships among these indices.

### 3.2.3 TRAINING STRATEGY AND BEYOND

In alignment with previous pre-training paradigms, we harness self-supervised learning to train our fMRI foundation model, leveraging the reconstruction of input fMRI signals. While we share a kinship with transformer-based models in the pursuit of modeling long-range dependencies, our proposed fMRI-PTE distinguishes itself through its unique merits and innovations.

Our key contribution lies in the design of a two-stage autoencoder, strategically engineered to expedite model convergence and introduce a unified foundation model tailored to cross-individual fMRI signals. Unlike VQGAN (Esser et al., 2021), fMRI-PTE possesses the capability to compress fMRI signals into a compact low-dimensional space (e.g., 1 dimension), thereby serving as a pre-trained model primed for enhancing a spectrum of downstream tasks, such as brain activity decoding.

In contrast to models like MAE (He et al., 2022) and MBM (Chen et al., 2022), our approach retains the dual functionality of both encoder and decoder. This duality affords us the versatility to employ the model for feature representation learning from cross-individual fMRI signals via the encoder or for the generation of surface images of fMRI signals.

## 4 EXPERIMENTAL RESULTS

### 4.1 DATASETS AND SETTINGS

**UK Biobank (UKB) (Miller et al., 2016).** It is a large-scale biomedical database and research resource, containing in-depth genetic and health information from half a million UK participants. We use the resting-state fMRI from UKB as pre-training dataset. Our study made use of pre-processed image data generated by an image-processing pipeline developed and run on behalf of UKB(Alfaro-Almagro et al., 2018). There are about $39,630$ subjects used for training and $1,000$ subjects held out for validation and every subject has one session of $490$ volumes.

**Natural Scenes Dataset (NSD) (Allen et al., 2022).** It is collected from 8 subjects, each of which is asked to view complex natural images from COCO dataset (Lin et al., 2014). We use fMRI-image pairs from NSD to evaluate models. We averaged repetitive fMRI trials of the same images. The training dataset was curated by utilizing distinct visual stimuli specific to individual subjects, while the testing dataset consisted of stimuli that were shared across all subjects (*i.e.*, 982 images). We evaluate models using the testing sets from subjects 1, 2, 5 and 7.

Table 1: **Quantitative analysis of cross-subject fMRI Reconstruction.** All models are trained on UKB dataset and directly evaluated on four subjects of NSD (*i.e.*, 1, 2, 5 and 7). Average results are reported. The subscript represents the dimension of the compression feature.

| METHOD | Pearson's ↑ | SSIM ↑ | MSE ↓ |
|---|---|---|---|
| MAE (He et al., 2022) | $0.8439 \pm 0.006$ | $0.6642 \pm 0.018$ | $0.0314 \pm 0.001$ |
| MBM (Chen et al., 2022) | $0.7372 \pm 0.016$ | $0.6229 \pm 0.010$ | $0.0699 \pm 0.002$ |
| LEA (Qian et al., 2023) | $0.8524 \pm 0.005$ | $\mathbf{0.8098 \pm 0.009}$ | $0.0296 \pm 0.001$ |
| Upper Bound | $0.8547 \pm 0.004$ | $0.7938 \pm 0.005$ | $0.0262 \pm 0.001$ |
| fMRI-PTE | $\mathbf{0.8546 \pm 0.004}$ | $0.7937 \pm 0.006$ | $\mathbf{0.0263 \pm 0.001}$ |

To investigate the efficacy of fMRI foundation models, we first measure the quality of reconstructed surface images with inputs. The quantitative metrics include Peasrson Correlation Coefficient (Pearson's) (Cohen et al., 2009), Structural Similarity Index Metric (SSIM) (Wang et al., 2004) and Mean Square Error (MSE). Then, we use the extracted features from fMRI foundation model to facilitate the downstream task of brain decoding. Similar to Brain-Diffuser (Ozcelik & VanRullen, 2023), we present several low-level and high-level metrics to measure the fidelity and accuracy of the generated images from fMRI signals. Concretely, the low-level metrics focus on the similarity of shallow features and contours (*e.g.*, PixCorr, SSIM and AlexNet), while the high-level metrics leverage deep features from CLIP (Radford et al., 2021) and Inception (Szegedy et al., 2016).

**Implemental Details**. All experiments are implemented with the PyTorch toolkit. For the first stage of reconstruction, the encoder $\mathcal{E}_1$ and decoder $\mathcal{D}_1$ have the symmetric structure with 5 blocks, containing the same $[1, 1, 2, 2, 4]$ residual layers as (Esser et al., 2021). We initialize them with pre-trained weights from ImageNet datasets (Deng et al., 2009). During training, we adopt Adam optimizer with $\beta_1 = 0.5$, $\beta_2 = 0.9$. The initial learning rate is set to $3e - 5$. We manually drop the learning rate by half once, and the total training iteration is about 600K. For the second stage of compression, both encoder $\mathcal{E}_2$ and decoder $\mathcal{D}_2$ have stacks of transformer blocks to improve the capacity. They are designed with the depth of 24, 2048 feature dimensions and 16 multi-heads in self-attention layers. We instantiate the mapper layer with two fully-connected layer, to compress the feature dimension from $257 \times 1024$ to $77 \times 768$. AdamW optimizer with hyperparameters $\beta_1 = 0.9$, $\beta_1 = 0.95$, and a batch size of 384 are used for training. We apply a linear learning rate schedule to first rain the model from scratch with $10,000$ training subjects from UKB dataset for 200K iterations. Then, we fine-tune it with all training subjects for the remaining 600K iterations, which takes about 6 days with 8 NVIDIA A100 GPUs. During inference, we can successfully extract well-represented features for arbitrary fMRI signals across different individuals. The compressed features can be further aligned with semantic information (*e.g.*, image features) to perform brain activity decoding. Moreover, our encoder can adapt to either stable diffusion (SD) (Rombach et al., 2022) or MaskGIT (MG) (Chang et al., 2022) to decode.

## 4.2 CROSS-SUBJECT EVALUATION

We initially assess the effectiveness of our fMRI-PTE across a diverse set of subjects, presenting a novel benchmark that holds greater significance and presents heightened challenges compared to within-subject evaluations. Achieving robust performance necessitates models to proficiently decode valid semantic information from fMRI signals, transcending the mere fitting of data distributions from a single individual. In this context, we engage in two typical experiments, encompassing cross-subject fMRI reconstruction and decoding.

### 4.2.1 FMRI RECONSTRUCTION AND ABLATION

**Settings and Baselines.** To assess the quality of our reconstructed images, we benchmark our model against several self-supervised methods, which include a masked autoencoder (MAE) (He et al., 2022), an fMRI pre-training model (MBM) (Chen et al., 2022), and an fMRI reconstruction model (LEA) (Qian et al., 2023). To ensure equitable comparisons, we made slight adjustments to these baseline models to achieve a similar number of parameters. All models underwent training on the UKB dataset for an equivalent number of iterations before being directly evaluated on four subjects from the NSD dataset.

**Results Analysis.** In Table 1, we present the averaged results from four subjects. The upper bound in the table reflects the remarkable outcomes achieved by our initial stage reconstruction autoen-

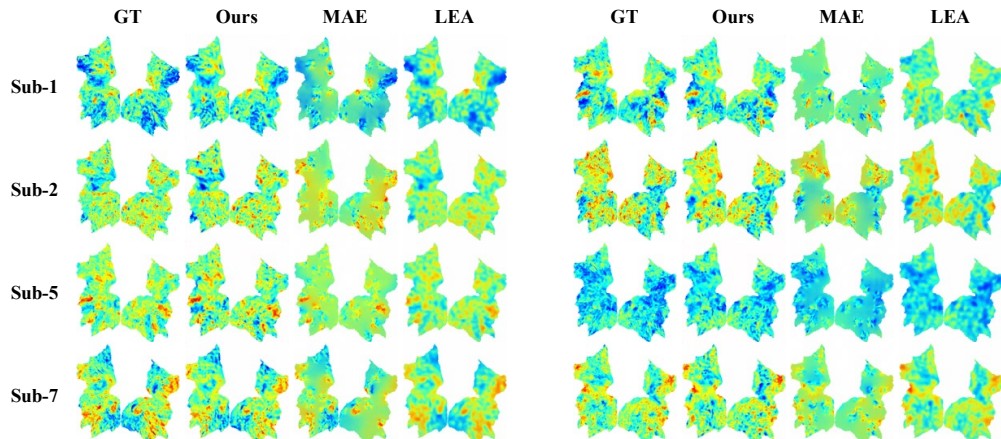

Figure 3: Visualizations of cross-subject fMRI reconstruction on NSD dataset.

Table 2: **Quantitative results of one-to-one cross-subject brain decoding on NSD dataset.** All models are trained on one individual and then tested on another. 'LR' represents the baseline of linear regression.

| METHOD | $7 \to 1$ | | $5 \to 2$ | | $1 \to 5$ | | $2 \to 7$ | |
|---|---|---|---|---|---|---|---|---|
| | AlexNet ↑ | CLIP ↑ | AlexNet ↑ | CLIP ↑ | AlexNet ↑ | CLIP ↑ | AlexNet ↑ | CLIP ↑ |
| MinD-Vis (Chen et al., 2022) | 59.67% | 56.71% | 62.47% | 57.03% | 59.96% | 55.71% | 62.56 % | 57.06% |
| LEA (Qian et al., 2023) | 70.39% | 60.89% | 70.61% | 61.75% | 68.47% | 63.80% | 68.89% | 61.45% |
| LR (Gifford et al., 2023) | 62.42% | 59.62% | 58.45% | 57.21% | 65.49% | 61.99% | 60.70% | 57.87% |
| fMRI-PTE(SD) | 64.15% | **63.68%** | 64.19% | 62.84% | 68.36% | **65.71%** | 65.46% | **62.73%** |
| fMRI-PTE (MG) | **72.82%** | 63.21% | **73.08%** | **63.44%** | **70.05%** | 65.47% | **70.36%** | 62.56% |

coder. Our innovative two-stage learning approach empowers our fMRI-PTE method to consistently outperform competitors across all three key metrics. Impressively, this superior performance is achieved alongside a remarkable compression ratio of 78%, effectively reducing feature dimensions from $257 \times 2048$ to $77 \times 768$. It is noteworthy that both MAE and MBM, which employ a similar masked autoencoding strategy, yield less favorable results. This divergence could be attributed to the reduced spatial redundancy inherent in brain surface images compared to natural images, necessitating more extensive training efforts. Comprehensive quantitative results are visualized in Fig. 3 and Fig. 7 (bigger figure in Appendix). While LEA exhibits a slight advantage over our method in terms of SSIM, it tends to generate blurrier images, primarily emphasizing low-frequency signals. This distinction is prominently depicted in Fig. 7 (Appendix), highlighting our findings.

### 4.2.2 BRAIN ACTIVITY DECODING AND ABLATION

**Settings and Baselines.** Previous studies are principally based on single individual evaluation, thus fail to perform cross-subject brain decoding. For a fair comparison, we take MinD-Vis (Chen et al., 2022) and LEA (Qian et al., 2023) as our primary baselines and re-train them by replacing fMRI vectors with our pre-processed surface images. Both baselines are published recently and only use the vision modality. Two settings are considered here for evaluation: **(1) one-to-one:** models trained on each individual are tested on the corresponding testing set; **(2) many-to-one:** Samples from several subjects are used to align fMRI and image features only. Once done, models trained on UKB dataset are directly tested on the testing set from one unknown subject.

**Result Analysis.** We provide a comprehensive analysis of our results in Tables 2 and 3, Figures 6 and 5. Several noteworthy observations emerge: (1) Our fMRI-PTE method excels in terms of ensuring the semantic consistency of decoded stimuli images. (2) Despite employing masked autoencoding for fMRI signals and leveraging powerful diffusion models for image generation, MinD-Vis experiences a pronounced performance decline, as clearly indicated in Table 2. This outcome is entirely understable, as MinD-Vis was not originally designed for the cross-subject task at hand. In contrast, fMRI-PTE showcases exceptional generalization prowess. Notably, for the second setting, MinD-Vis is omitted from the comparison in Table 3, highlighting its constrained utility in this context. (3) LEA, employing a comparable pipeline with our approach, utilizing ridge regression for alignment, falls behind significantly, highlighting the effectiveness of our fMRI pre-trained encoder. (4)

Table 3: **Quantitative results of many-to-one cross-subject brain decoding on NSD dataset.** All models are trained on UKB dataset. '257 → 1' denotes we use fMRI-image pairs of three subjects (Sub-2, Sub-5 and Sub-7) from NSD dataset to align features of fMRI and images, and so on for other settings. It is implemented by ridge regression for all models, and do not tune other parameters. 'LR' represents the baseline of linear regression.

| METHOD | 257 → 1 | | 157 → 2 | | 127 → 5 | | 125 → 7 | |
|---|---|---|---|---|---|---|---|---|
| | AlexNet ↑ | CLIP ↑ | AlexNet ↑ | CLIP ↑ | AlexNet ↑ | CLIP ↑ | AlexNet ↑ | CLIP ↑ |
| LEA (Qian et al., 2023) | 72.32% | 64.52% | 71.96% | 65.01% | 68.17% | 65.97% | 69.80% | 64.21% |
| LR (Gifford et al., 2023) | 67.04% | 63.34% | 66.26% | 64.03% | 66.02% | 63.12% | 62.45% | 60.77% |
| fMRI-PTE (SD) | 70.95% | **68.96%** | 69.95% | **69.08%** | 69.13% | **67.77%** | 68.28% | **65.93%** |
| fMRI-PTE (MG) | **76.66%** | 66.83% | **76.45%** | 68.15% | **72.88%** | 67.46% | **72.63%** | 65.30% |

Table 4: **Quantitative analysis of within-subject brain decoding on NSD dataset.** Four subjects (Sub-1, Sub-2, Sub-5 and Sub-7) are selected as the testing set. Models are trained and tested on each individual. 'LR' represents the baseline of linear regression. Average results are reported.

| METHOD | M | Low-Level | | | | High-Level | | | |
|---|---|---|---|---|---|---|---|---|---|
| | | PixCorr ↑ | SSIM ↑ | AlexNet(2) ↑ | AlexNet(5) ↑ | Inception ↑ | CLIP ↑ | EffNet-B ↓ | SwAV ↓ |
| Mind-Reader | V&T | - | - | - | - | 78.20% | - | - | - |
| SD-Brain | V&T | - | - | 83.00% | 83.00% | 76.00% | 77.00% | - | - |
| BrainCLIP | V&T | - | - | - | - | 79.70% | 89.40% | - | - |
| Cortex2Image | V | **.150** | **.325** | - | - | - | - | .862 | .465 |
| MinD-Vis | V | .080 | .220 | 72.12% | 83.15% | 78.77% | 76.17% | .854 | .491 |
| LEA | V | .119 | .111 | 77.41% | 87.36% | 82.31% | 80.80% | .851 | .445 |
| BrainCLIP | V | - | - | - | - | 73.30% | **83.60%** | - | - |
| LR | V | .060 | .250 | 69.05% | 78.72% | 76.04% | 79.01% | .863 | .524 |
| fMRI-PTE (SD) | T | .067 | .257 | 71.67% | 83.22% | 80.29% | 82.02% | .852 | .513 |
| fMRI-PTE (MG) | V | .131 | .112 | **78.13%** | **88.59%** | **84.09%** | 82.26% | **.837** | **.434** |

Lastly, we present 'LR' as a baseline using linear regression, where we serialize pixels of the surface map (excluding the background) as vectors of fMRI signals and regress them to the image space for brain decoding. This can be seen as a variant of our fMRI-PTE, albeit without the fMRI pre-trained encoder. The results from both tables collectively emphasize the clear superiority of our proposed methodology.

## 4.3 WITHIN-SUBJECT EVALUATION

**Settings and Baselines.** As a standard setting for brain activity decoding, models are trained individually on each subject. We compare our methods with several competitors, including Mind-Reader (Lin et al., 2022), SD-Brain (Takagi & Nishimoto, 2023), Cortex2Image (Gu et al., 2022), MinD-Vis (Chen et al., 2022), LEA (Qian et al., 2023), BrainCLIP(Liu et al., 2023). Among them, we primarily conduct discussions with methods that only use the vision modality.

**Result Analysis.** Table 4 reports averaged results of four testing subjects on NSD dataset. First, fMRI-PTE achieves remarkable performance with regard to both low-level and high-level metrics. Some of them even surpass methods which additionally use the text modality. It significantly demonstrates the effectiveness and universality of our method. Second, our model achieves higher performance than LEA and LR on most of metrics, which clearly indicates the ability of our method to extract discriminative features for fMRI signals. Third, we visualize decoded images in Fig. 4 (Appendix), further supporting the quantitative results and conclusions described above.

## 5 CONCLUSION

In this paper, we introduce fMRI-PTE, a specialized tool designed for fMRI signals. It tackles the challenge of varying data dimensions caused by individual differences in brain structure. By converting fMRI signals into a consistent 2D format and using a unique two-step learning approach, we improve the quality of fMRI pre-training. This flexibility, combined with image generators, results in well-defined fMRI features. These features make it easier to decode brain activity within and across individuals. Our contributions, such as data transformation and efficient training methods, have been thoroughly tested in experiments. The positive results provide a strong basis for further exploration in this field.

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

# 6 APPENDIX

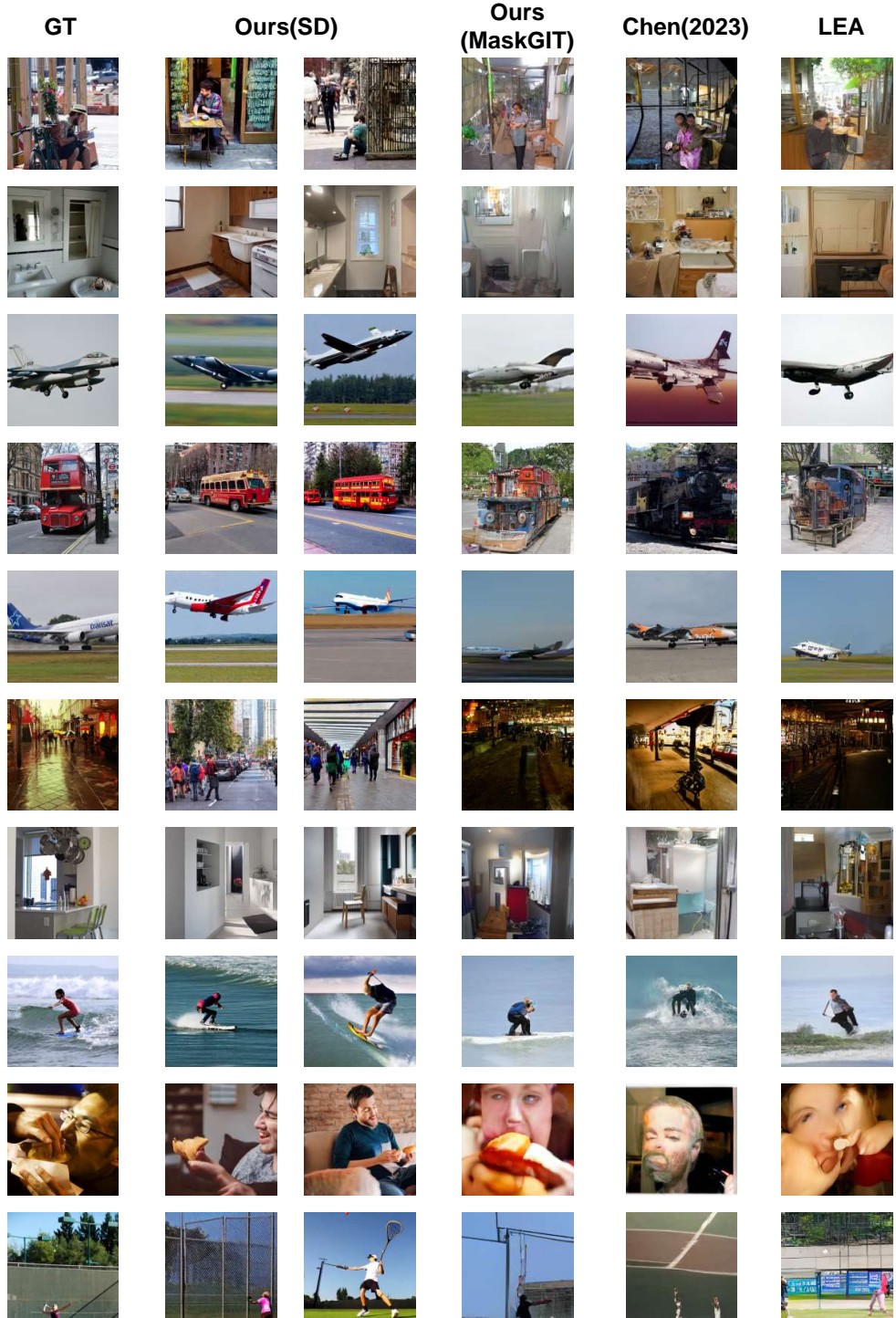

Figure 4: Visualizations of within-subject fMRI reconstruction on NSD dataset.

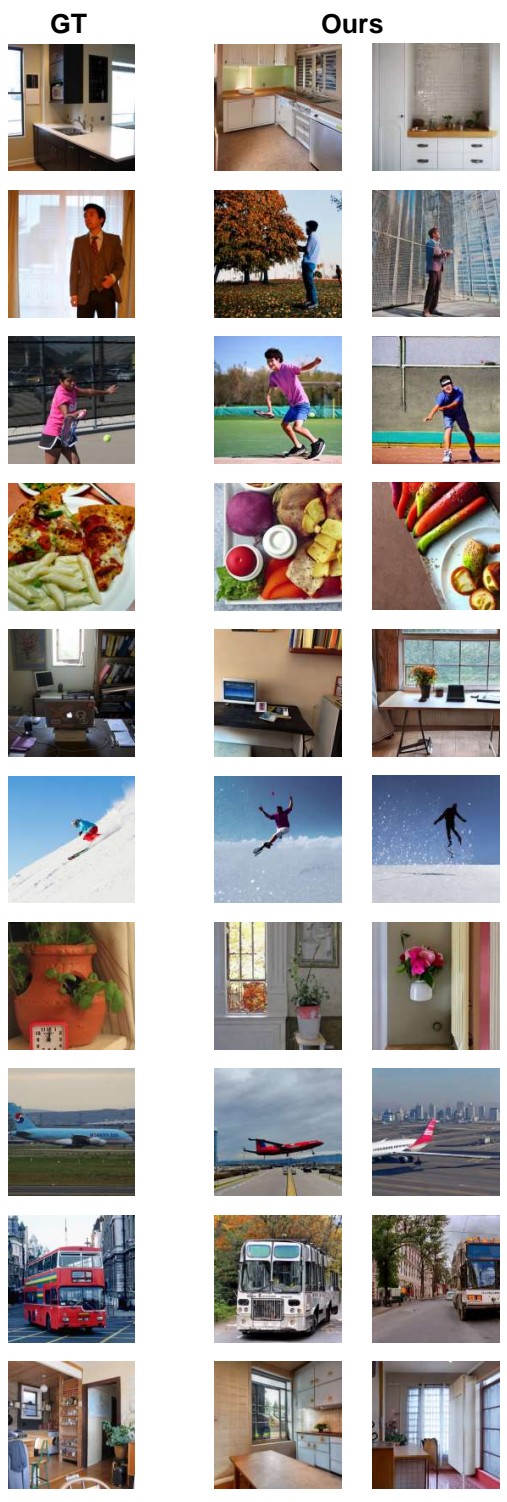

Figure 5: Visualizations of many-to-one cross-subject fMRI reconstruction on NSD dataset.

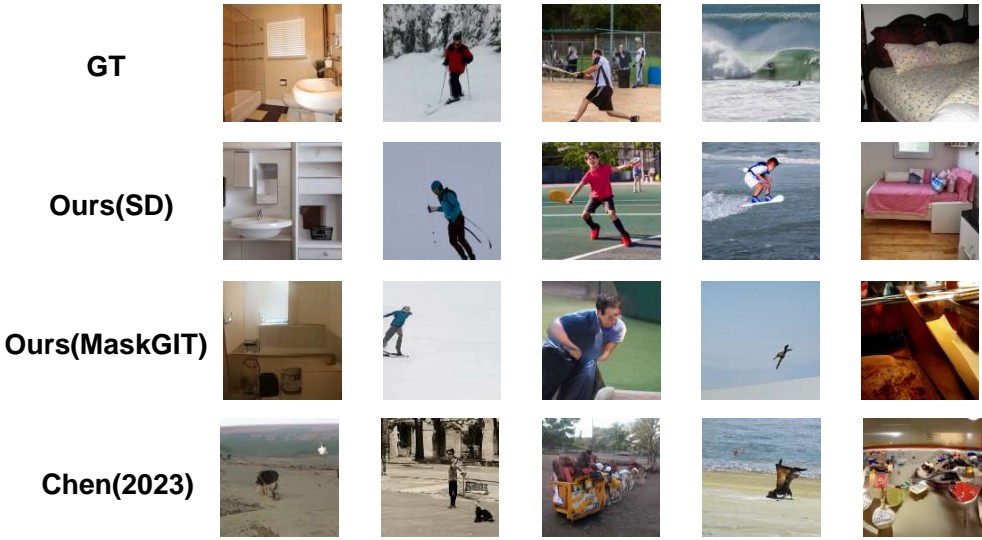

Figure 6: Visualizations of one-to-one cross-subject fMRI reconstruction on NSD dataset.

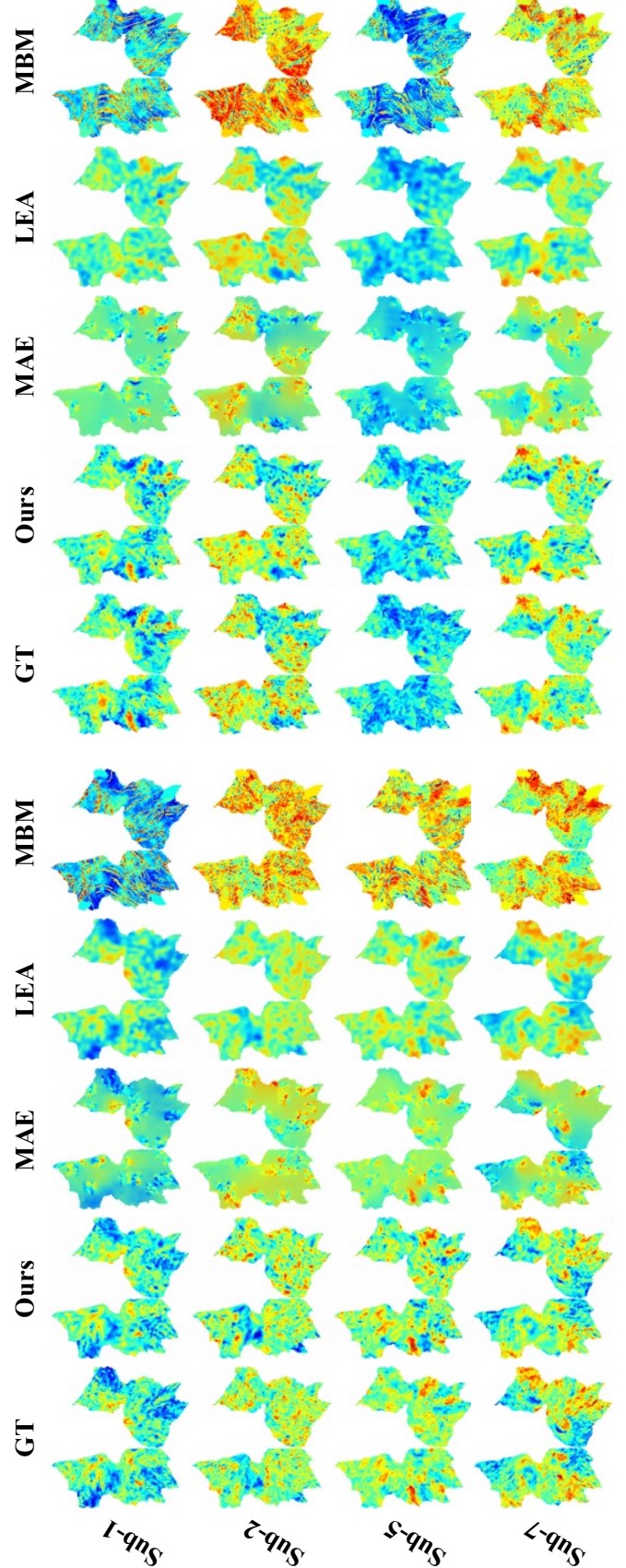

Figure 7: Bigger size of Visualizations of cross-subject fMRI reconstruction on NSD dataset.

