# OpenReview forum: "fMRI-PTE: A Large-scale fMRI Pretrained Transformer Encoder for Multi-Subject Brain Activity Decoding"
_ICLR.cc/2024/Conference — Submitted to ICLR 2024_

### Official Review · Reviewer_MXbE · 2023-10-26

**Soundness:** 1 poor
**Presentation:** 2 fair
**Contribution:** 1 poor
**Rating:** 3
**Confidence:** 4

**Summary:**

In this paper, the authors train an autoencoder architecture in a self-supervised way on a large dataset of fMRI recordings (UK BioBank) in order to create a general model that could be readily used to solve downstream neuroinformatic tasks. They first test the reconstruction capability of their method, achieving results (in terms of correlation, SSIM and MSE) comparable to those obtained with alternative methods. They then test the proposed method using an image decoding task based on the Natural Scenes Dataset (NSD) and compare its performance with other possible architectures (e.g., MAE, LEA) using a variety of evaluation metrics, reporting favorable results.

**Strengths:**

-	The article is generally readable, and the research work is well-motivated.
-	Building large-scale foundation models for fMRI data would constitute a valuable asset to more effectively process neuroimaging data.

**Weaknesses:**

-	Though the manuscript is readable, the research goals are often unclear or framed in a general way. For example, the title puts emphasis on “brain activity”, but in fact the results only refer to “cortical activity” (surface data), and the main results more specifically focus on a visual decoding task.
-	There are several methodological details that require clarification (see questions below).
-	The proposed architecture involves the combination of several modules and training phases; more quantitative analyses (e.g., by means of ablation studies) should be performed to justify each design choice.

**Questions:**

-	The authors should more clearly explain how their “fMRI-PTE” architecture differs from masked auto-encoder architectures used in the deep learning literature and from recent architectures specifically introduced to model neuroimaging data. In general, the authors use vague sentences such as “our proposed fMRI-PTE distinguishes itself through its unique merits and innovations” that do not allow to clearly understand why and how their two-stage architecture should work better than existing (and possibly simpler) ones. More systematic ablation studies on the fMRI-PTE architecture would help clarify this.
-	The authors should similarly explain how their “innovative approach to transform fMRI signals into unified representations by projecting them onto 2D brain activation images” differs from standard methods that allow mapping 3D fMRI volumes into 2D surface (cortical) representations. From the authors’ description, it seems they are just adopting standard preprocessing pipelines that convert MNI volumes to surface activations.
-	I do not agree that the pipeline in Fig. 1 is self-explanatory and meticulously depicted. The caption should at least explain the meaning of each model box, the reason why pre-processing using “downstream data” precedes the training of the foundation model, the meaning of the “fires” and “iced” symbols, etc. etc. Also, why is there a block named “GPT”? Such a model is never mentioned in the manuscript.
-	More details about the specific vector quantization approach are required to fully understand the feature extraction process. What happens if the features are not quantized? This should also be inspected by ablation studies.
-	The role of each loss term (perceptual, adversarial, reconstruction, and commitment) should be investigated more in detail to establish whether such combination is really required to improve performance.
-	Why is a Vision Transformer required to compress the quantized indices? It would be useful to also investigate simpler autoencoder architectures.
-	Table 1: “The subscript represents the dimension of the compression feature.” I do not see any subscript in the table. Furthermore, the differences with respect to existing methods seem quite marginal.
-	For the decoding task, the authors mention that they selected a subset of Visual ROIs. However, from the current description it is not clear how such subset of the surface data was given as input to the pre-trained model (which, from my understanding, expects as input the entire cortical image).
-	The images in Fig. 3 suggest that the input is just the subset of Visual ROIs shown in Fig. 2. However, I am a bit puzzled by the overall poor reconstruction accuracy of MAE, which suggests that such architecture has been trained in a sub-optimal fashion.
-	“Implemental Details” should be described in the Methodological section, not in the Results section.
-	English phrasing and grammar could be improved. Just as an example, in the abstract the authors write that they introduce a novel learning strategy “tailored for pre-training 2D fMRI images”. It’s not the images that are pre-trained, but rather a model of these images. I strongly recommend the authors to carefully check the entire document to improve the construction of sentences, at the same time avoiding unnecessary repetitions (for example, there is also a little abuse of the terms “intricacies” and “intricate”, which are repeated several times, often within the same paragraph).
-	Some acronyms are used before being explicitly introduced.

---

> ### Author Response · Authors · 2023-11-23
> **Response to Reviewer MXbE**
>
> Thanks for your time reviewing and providing professional feedback for our manuscript. Our codes and models will be released, and here are responses to specific questions.
>
> **1. The research goals are often unclear or framed in a general way**
>
> Thanks and we apologize for confusions. Following previous literature [A,B], we use the same word of 'brain activity decoding' to define our task. Also, we focus on visual cortical regions for a fair comparison. As for surface data, it is one of our contributions. Different from previous work that directly uses flattened 3D voxels of fMRI data as input, we additionally transform 3D voxels to 2D surface images and take them as a good carrier of information.
>
> **2. More quantitative analyses should be performed**
>
> Thanks for raising this question. As indicated by the title of subsections, our ablation studies are discussed in Sec, 4.2.1 and 4.2.2, as well as Tab. 1. More concretely, 'LR' and 'LEA' are two important baselines, where the former excludes the pre-trained encoder and the latter only has one stage for fMRI pre-training. Experiments in Tabs. 1, 2 and 3 clearly demonstrate the effectiveness of our proposed modules.
>
> **3. How fMRI-PTE differs from masked auto-encoder architectures?**
>
> Thanks, we explain that our method and MAE both use autoencoders for pre-training. Differently, we reconstruct the original images from the compressed features, but MAE adopts the masked modeling strategy. To verify the efficacy of two-stage architecture, we compare with LEA, a competitor that only takes one-stage architecture for reconstruction. Moreover, we make a slight modification to LEA to ensure a similar number of parameters and training iterations. Under this circumstance, results in Tabs. 1~4 clearly demonstrate the superiority of our design.
>
> **4. Innovative approach to transform fMRI signals**
>
> We apologize for any misleading. As explained in Sec. 3.1, we adopt the off-the-shelf toolkit to transform 3D voxels of fMRI data to 2D surface images. We acknowledge that it is a common preprocessing in neuroscience. However, taken it as a good carrier of information and further explore brain decoding on it is novel. Compared with previous method that mostly takes 1D vectors of voxels as input, experiments in Sec. 4.2 and 4.3 have demonstrated the effectiveness of this representation of fMRI data. We will revise the description and tone down the claim, making them more precise.
>
> **5. Pipeline in Fig. 1 is not meticulously depicted**
>
> Thanks for your suggestion. We will polish the figure and add more descriptions. For your specific questions, (1) during pre-training, we use fMRI data only from UKB dataset. (2) GPT refers to decoder-only transformers, which serves as the decoder $\mathcal{D}_{2}$ in Sec. 3.2.2. (3) the 'fires' and 'ices' symbols represent whether model parameters are learned or fixed during training.
>
> **6. More details about vector quantization and corresponding loss terms**
>
> Thanks for the question. As stated in Sec. 3.2.1, the architecture and training objectives are the same as VQGAN [C]. Thus, we refer readers to [C] for more details. We will add the footnote to make it more clear in the revision.
>
> **7. Why is a Vision Transformer required to compress the quantized indices?**
>
> Thanks. Without feature compression, the autoencoder performs more like an identity mapping, passing 100% information from the encoder to the decoder. It would significantly improve the quality of reconstruction, but not help the encoder's learning.
>
> **8. Is input the entire cortical image?**
>
> No, our input for both pre-training and brain decoding task is surface image only with visual ROIs. Concretely, we transform the whole cortex to 2D images, and use the mask of visual RoIs to crop it. We will make it more clear in the revision.
>
> **9. Poor reconstruction accuracy of MAE?**
>
> Good comments. As explained in Sec. 4.2.1, we are also surprised by the poor reconstruction accuracy of MAE. We explain that possible reasons could be (1) less spatial redundancy in brain surface images compared to natural images; (2) large white background in brain surface images (note that our method also reconstructs the background); (3) more training efforts required for masked modeling strategy. In contrast, our method is tailored to brain surface images, showing its superiority.
>
> **10. Writing, acronyms and Typos**
>
> We apologize for any mistake. We will correct them and do careful proofreading multiple times in the revision.
>
>
>
>
> **References**
>
> [A] Takagi, Yu, and Shinji Nishimoto. "High-resolution image reconstruction with latent diffusion models from human brain activity." CVPR, 2023
>
> [B] Chen, Zijiao, Jiaxin Qing, and Juan Helen Zhou. "Cinematic Mindscapes: High-quality Video Reconstruction from Brain Activity." NeurIPS, 2023
>
> [C] Patrick Esser, Robin Rombach, and Bjorn Ommer. "Taming Transformers for High-resolution Image Synthesis". CVPR2022

---

### Official Review · Reviewer_JDw2 · 2023-11-01

**Soundness:** 2 fair
**Presentation:** 2 fair
**Contribution:** 2 fair
**Rating:** 3
**Confidence:** 3

**Summary:**

The manuscript proposes an autoencoder-based approach (fMRI-PTE) to brain decoding. The experiments show improved fMRI reconstruction performance and better brain decoding results on the NSD dataset across multiple metrics compared to previous baselines. The VQGAN approach for image synthesis inspired the proposed architecture.

**Strengths:**

- Multiple baselines
- Multiple experiments for one upstream and one downstream task.

**Weaknesses:**

- The manuscript lacks a strong baseline for the NSD dataset: MindEye (Scotti et al., 2023).
- The foundational model and universal applicability claims are not well supported, as the manuscript has not explored the model's scaling capabilities. It has been trained and evaluated on two datasets and one downstream task (brain decoding), while there are other tasks in brain studies. Furthermore, no zero-shot capabilities have been explored. Lastly, no analysis has been performed to validate whether the performance is reliable across demographics.
- Innovative Data Transformation is not novel.
   - It is a known procedure. Otherwise, the 32k_fs_LR toolboxes will not be available (https://netneurolab.github.io/neuromaps/index.html, https://github.com/DiedrichsenLab/fs_LR_32).
   - The better-performing MindEye (Scotti et al., 2023) used voxels. But there are other ways to transform 4D fMRI to 2D data by using DiFuMo (Dadi et al., 2020), Neuromark (Du et al., 2020), or Spectral Clustering (Geenjaar et al., 2022). These methodologies could be ablated.
- Claims about preserving high-frequency signals are confusing and are not discussed with corresponding literature. Note each time point is acquired for 1.5 seconds at UKBioBank. The fMRI is highly undersampled and, most importantly, noisy. It is not that easy in general (Trapp et al., 2018). Hence, the manuscript needs to be more specific about that. For evaluation, you can ensure that you reconstruct frequency modes (e.g., Yuen et al., 2019). You can also consider the works from the speech domain to ensure frequency reconstructions (Kumar et al., 2023; Yamamoto et al., 2020).
- Additionally, you need to validate the claim about spatial interaction across brain regions. An evaluation could ensure the static/dynamic functional connectivity is preserved after reconstruction.


- References:
  - Dadi, Kamalaker, et al. "Fine-grain atlases of functional modes for fMRI analysis." NeuroImage 221 (2020): 117126. https://parietal-inria.github.io/DiFuMo/
  - Du, Yuhui, et al. "NeuroMark: An automated and adaptive ICA based pipeline to identify reproducible fMRI markers of brain disorders." NeuroImage: Clinical 28 (2020): 102375.
  - Geenjaar, Eloy, et al. "Spatio-temporally separable non-linear latent factor learning: an application to somatomotor cortex fMRI data." arXiv preprint arXiv:2205.13640 (2022).
   - Trapp, Cameron, Kishore Vakamudi, and Stefan Posse. "On the detection of high-frequency correlations in resting state fMRI." Neuroimage 164 (2018): 202-213.
   - Yuen, Nicole H., Nathaniel Osachoff, and J. Jean Chen. "Intrinsic frequencies of the resting-state fMRI signal: the frequency dependence of functional connectivity and the effect of mode mixing." Frontiers in Neuroscience 13 (2019): 900.
   - Kumar, Rithesh, et al. "High-Fidelity Audio Compression with Improved RVQGAN." arXiv preprint arXiv:2306.06546 (2023).
   - Yamamoto, Ryuichi, Eunwoo Song, and Jae-Min Kim. "Parallel WaveGAN: A fast waveform generation model based on generative adversarial networks with a multi-resolution spectrogram." ICASSP 2020-2020 IEEE International Conference on Acoustics, Speech and Signal Processing (ICASSP). IEEE, 2020.
    - Scotti, Paul S., et al. "Reconstructing the Mind's Eye: fMRI-to-Image with Contrastive Learning and Diffusion Priors." arXiv preprint arXiv:2305.18274 (2023).

**Questions:**

- The MAE baseline is confusing and not comparable. In MAE, they use lighter decoders, and improving reconstruction did not improve the downstream performance; hence, they did not focus on reconstruction. In addition, a higher masking ratio leads to situations where the objects are not reconstructed.
- I did not find details for the preprocessing of the fMRI dataset since there is no appendix. Have you applied motion correction and aligned the fMRI to the reference volume?

Rigor:
- Lack of variability (STD, SE, or IQR) in Table 2, Table 3, and Table 4.
- Lack of statistical analysis to compare models.

Clarity:
- The text is too wordy and not specific enough.

---

> ### Author Response · Authors · 2023-11-23
> **Response to Reviewer JDw2**
>
> Thanks for your time reviewing and providing professional feedback for our manuscript. Our codes and models will be released, and here are responses to specific questions.
>
>
> **1. Lacks a strong baseline of MindEye**
>
> Thanks. The main purpose of this paper is for cross-subject brain activity decoding. Experiments in Tab. 2 and 3 have demonstrated the advantage of our method. Like previous methods, MindEye is for within-subject brain decoding, and it needs to train from scratch for each subject. However, it is a good work that utilizes diffusion prior for fMRI-Image mapping, published in arXiv before the submission. We will cite and discuss it in the revision.
>
> **2. No zero-shot capabilities have been explore**
>
> Thanks for the question. We would like to clarify that,
>  * Our proposed fMRI pre-trained transformer encoder is mainly for cross-subject brain activity decoding. It would be interesting future work to explore its application to other tasks, like brain disease classification.
>  * In addition to cross-subject brain decoding task, we also reported results on within-subject task, achieving competitive or even better performance compared with end-to-end training competitors. It clearly suggests the generalization ability of our proposed method.
>  * Results in Tab. 2 and 3 exactly show the zero-shot capability of our method, where we use data from one or many subjects to align fMRI and image features, and directly evaluate the model on one unknown subject. The higher performance achieved by our method indicates its superiority
>
> **3. Innovative Data Transformation is not novel**
>
> We apologize for any misleading. As explained in Sec. 3.1, we adopt Connectome Workbench and pycortex toolboxes to transform 3D voxels of fMRI data to 2D surface images. We acknowledge that it is a common preprocessing in neuroscience. However, taken it as a good carrier of information and further explore brain decoding on it is novel. Compared with previous method that mostly takes 1D vectors of voxels as input, experiments in Sec. 4.2 and 4.3 have demonstrated the effectiveness of this representation of fMRI data. We will revise the description and tone down the claim, making them more precise.
>
> **4. Claims about preserving high-frequency signals are confusing**
>
> Thanks, but we would like to clarify some misunderstandings that,
>  * Low- and high-frequency refer to signals in 2D images, because we transform 3D voxels of fMRI data to 2D surface images.
>  * We use the autoencoder to reconstruct 2D surface images frame by frame, instead of performing the temporal reconstruction
>  * Based on the above clarification, we have evaluated the quality of reconstructed surface images with three image-based metrics in Tab. 1, showing the advantage of our method.
>
> **5. The MAE baseline is confusing and not comparable**
>
> Thanks, but we would like to clarify that,
>  * In addition to MAE, we also compare our method with two competitors, MBM and LEA, that specifically designed autoencoder for fMRI data.
>  * A higher masking ratio is the default setting of masked modeling strategy. Notably, MBM also uses the similar masked modeling strategy with a higher masking ratio.
>  * For a fair comparison, we made a slight adjustments to these baseline models to ensure a similar number of parameters and training iterations.
>
> **6. Details for the preprocessing of the fMRI dataset**
>
> Thanks. The UKB and NSD datasets used in our paper were already preprocessed into MNI and fsaverage group space and we only applied transformation between group spaces. According to the descriptions of UKB and NSD, they did apply motion correction.
>
> All of these clarifications have been explained in the first paragraph of Sec. 4.2.1, so results in Tab. 2 are enough to show the effectiveness of our method.
>
> **7. Lack of variability in Tabs. 2, 3 and 4**
>
> Thanks. As stated in Sec. 4.1, we follow [A] to use several low-level and high-level metrics to measure the fidelity and accuracy of the generated images from fMRI signals. As a result, there are totally 8 metrics used in Tabs. 2, 3, 4, which are comprehensive and enough to evaluate the effectiveness of our method.
>
> **References**
>
> [A] Ozcelik, Furkan, and Rufin VanRullen. "Brain-diffuser: Natural scene reconstruction from fmri signals using generative latent diffusion." arXiv preprint arXiv:2303.05334 (2023).

---

### Official Review · Reviewer_X3zc · 2023-11-05

**Soundness:** 3 good
**Presentation:** 2 fair
**Contribution:** 2 fair
**Rating:** 5
**Confidence:** 3

**Summary:**

This paper introduces a novel fMRI pre-training strategy, fMRI-PTE, for brain decoding. It entails the conversion method, which converts the volume or surface-wise fMRI image into a 2D image, and a self-supervised pre-training strategy that generates well-represented fMRI features. Additionally, this strategy facilitates various downstream tasks, including within-subject and cross-subject brain activity decoding.

**Strengths:**

- The authors have transformed fMRI signals into a 2D image format, potentially capturing low- and high-frequency signal variations as patterns. This method may also reduce computational costs compared to traditional approaches that linearize whole-brain voxels into 1D vectors.
- Employing the resting-state fMRI data for pre-training may enable the model to encompass a wide range of intrinsic individual brain information, which could potentially enhance the model’s performance on various downstream tasks.
- The study shows impressive reconstruction results.

**Weaknesses:**

- While the authors argue that their proposed method has its advantages over the existing methods, e.g., MBM, in the field, to this reviewer, it just exploits the existing methodologies for the problem of interest, fMRI-based brain decoding. In this regard, the methodological innovation is marginal, raising concern about its suitability for presentation in ICLR.
- Their “innovative data transformation” onto 2D brain activation images seems trivial. It uses the existing mapping method.
- There should be a more rigorous explanation and analysis of the contributions of their efficient training and learning strategy. To this reviewer, those are not clear.
- In Figure 1, the GPT model is included in both the ‘foundation model training’ and ‘downstream application’ parts. While the ViT is mentioned in ‘3.2.2 Transformer compression’ as being used in the encoder, the GPT is only mentioned in the introduction and lacks details in the method part.
- The ‘downstream application’ part in Figure 1 includes the ‘image decoder’ module. It seems to be an additional pre-trained model using natural images. However, the paper does not include sufficient details about it.
- It would be beneficial to provide information about the kernel size. Some brain regions, such as V6A and LO2, are quite small (Figure 3). If the kernel size is too big, it may be difficult to get the relationships between these small regions and other regions. This is because the small regions could potentially merge with other adjacent regions. Therefore, it would be better to specify the kernel size and its significance.
- In Table 2, the authors present the one-to-one cross-subject brain decoding results for specific pairs (i.e., 7 → 1, 5 → 2, 1 → 5, and 2 → 7), but the method for choosing these pairs is not clear. It would be informative to evaluate the results for one-to-one cross-subjects by examining the reverse directions or by calculating the average values for each source across all other target cases (e.g., from 7 to 1, 2, and 5, respectively) to determine the efficacy of the model/strategy for this task.

**Questions:**

- In the second sentence of the third paragraph under ‘3.2.2 Transformer Compression’, the two variables ‘L’ and ‘C’ are mentioned, which seem to be related to the number of tokens and feature dimensions, respectively. However, while the batch size is also mentioned, no corresponding variable seems to be provided.
- Presenting the brain visualizations with a color scale and matching their scales for the overall comparisons may improve the clarity of the results.

---

> ### Author Response · Authors · 2023-11-23
> **Response to Reviewer X3zc**
>
> Thanks for your time reviewing and providing professional feedback for our manuscript. Our codes and models will be released, and here are responses to specific questions.
>
> **1. The methodological innovation is marginal**
>
> Thanks for the question, but we would like to clarify some misunderstandings that,
>   * Both MBM and our method are proposed to build fMRI pre-trained autoencoder for brain decoding task, so they are directly comparable.
>   * Our method is not designed based on MBM. Though they have the same purpose of fMRI pre-training, they are quite different from two aspects, architecture and inputs. MBM takes as input 1D fMRI vectors, and use masked modeling [A] for pre-training. Instead, we take 2D fMRI surface images as inputs, and design two-stage learning strategy with dimensional compression for pre-training.
>   * Previous methods only focus on within-subject brain decoding. In this paper,  our technological contribution lies not only in the fMRI pre-trained autoencoder, but also in the framework for both within- and cross-subject brain decoding. Experiments in Sec. 4.2 and 4.3 have shown the efficacy of our method.
>
> **2. 'Innovative data transformation' seems trivial**
>
> We apologize for any misleading. As explained in Sec. 3.1, we adopt the off-the-shelf toolkit to transform 3D voxels of fMRI data to 2D surface images. We acknowledge that it is a common preprocessing in neuroscience. However, taken it as a good carrier of information and further explore brain decoding on it is novel. Experiments in Sec. 4.2 and 4.3 have demonstrated the effectiveness of this representation of fMRI data. We will revise the description and tone down the claim, making them more precise.
>
> **3. 'Analysis of efficient training and learning strategy**
>
> Thanks for raising the question. We would like to clarify that,
>  * The purpose of this paper is to design a fMRI pre-trained encoder for brain activity decoding. Thereby, the main experiments are within- and cross-subject brain decoding, which have been conducted and evaluated in Sec. 4.2 and 4.3.
>  * To further investigate the efficacy of our pre-trained encoder, we compare the quality of reconstructed images with other methods in Tab. 2. Specifically, we deliberately made slight adjustments to competitors to achieve a similar number of parameters and the same number training iterations, to verify the advantage of efficient training of our method.
>
> **4. Lacks details of GPT and image decoder in the method part**
>
> Thanks and we apologize for confusions. (1) GPT refers to decoder-only transformers, which serves as the decoder $\mathcal{D}_{2}$ in Sec. 3.2.2. (2) In the last paragraph of Sec. 4.1, we discuss that the image decoder could be either stable diffusion or MaskGIT. We also show results with both decoders in Tabs. 2~4. We will polish the figure and add more explanations.
>
> **5. It would be beneficial to provide information about the kernel size**
>
> Good question. The kernel size we used is $3\times 3$, and we resize the input surface images to $256\times 256$, so it could capture information in small regions. In this paper, we process fMRI data as 2D images, the setting of kernel size is standard in computer vision. So, we don't tune it heavily.
>
> **6. The method for choosing cross-subject pairs is not clear.**
>
> Thanks. We explain that the combination of cross-subject is numerous, especially for the one-to-one setting. In Tab. 2, we randomly choose 4 representative pairs for evaluation, where both the source and target domain cover 4 subjects. Combined with results in Tab. 3, experiments are self-contained and enough to show the capacity and superiority of our method on cross-subject brain decoding task. We will add more explanation about pairing choices in the revision.
>
> **7. Batch size is mentioned, but no corresponding variable seems to be provided**
>
> We sorry for typos. In Sec. 3.2.2., words of 'batch size' should be deleted. It is for simplicity of symbolic representation, and we explain the number of batch size in Sec. 4.1.
>
> **8. Presenting the brain visualizations with a color scale**
>
> Good suggestion. We post-process fMRI surface images by min-max normalization and visualize them in Fig. 3. We will further add colorbar to improve the clarity of results.
>
>
>
> **References**
>
> [A] He, Kaiming, Xinlei Chen, Saining Xie, Yanghao Li, Piotr Dollár, and Ross Girshick. "Masked autoencoders are scalable vision learners." CVPR2022

---

### Official Review · Reviewer_8D82 · 2023-11-06

**Soundness:** 3 good
**Presentation:** 2 fair
**Contribution:** 2 fair
**Rating:** 5
**Confidence:** 4

**Summary:**

The paper introduces fMRI-PTE, an auto-encoder-based pretraining framework for fMRI data that addresses the challenge of varying data dimensions across individuals by transforming brain signals into standardized 2D representations. This approach not only ensures dimensional consistency but also preserves the uniqueness of brain patterns, incorporating a new learning strategy that improves reconstruction quality for both within-subject and cross-subject decoding tasks. Validated by extensive experiments, fMRI-PTE stands out for its adaptability and universal application, presenting a significant step forward for research in brain activity analysis and its potential high-impact applications.

**Strengths:**

1. Innovative Use of Large Pretraining Datasets: The paper presents a promising and innovative approach by leveraging large pretraining datasets to enhance performance on smaller downstream tasks. This strategy is particularly interesting as it can potentially unlock new capabilities in machine learning models by exploiting the rich information present in expansive datasets to benefit tasks with limited data availability.
2. Effective Dimensionality Reduction Technique: The methodology for transforming fMRI signals into a uniform 2D format is a commendable strength of the paper.
3. Exploration of a Cutting-edge Application: The task of decoding visual signals from fMRI data is an engaging and cutting-edge application that stands out in the paper. The pursuit of this task is highly relevant to advancing the intersection of neuroscience and artificial intelligence, and it holds significant promise for future developments in brain-computer interfaces and medical diagnostic techniques.

**Weaknesses:**

1. Insufficient Empirical Evaluation: The paper needs to include some important ablation experiments to validate the proposed model. For example, the ablation study shows the effectiveness of pretraining on the UK Biobank datasets;
2. Unclear Methodology Description: The complexity of the training process is not adequately described in the current methodological section. The paper needs to include more detailed equations and procedural explanations to delineate the training steps precisely.
3. Presentation and Terminology Issues: The paper's presentation suffers from the use of abbreviations that are not clearly defined, which can lead to confusion and misinterpretation. (See in Question 1 and 5)
4. Lack of Supplementary Materials: For a paper detailing a complex training process, the absence of source code is a significant drawback. The provision of the actual code would greatly aid others in replicating the study. This support material is often crucial for peer reviewers and practitioners who wish to validate the claims or extend the work.

**Questions:**

1. "our methodology involved an initial conversion of GLM", please provide the full name of "GLM".
2. "Here, L and C represent batch size, the number of tokens, and feature dimensions." Why do two symbols, L and C, correspond with three definitions?
3. Can you explain what the input of "the encoder \Epsilon_2" is? Please provide symbols and feature dimensions to describe the input.
4. Can you provide the code and pre-train model to enlarge the work impact?
5. In Table 4, what is the meaning of column "M" and its value, "T" and "V".

---

> ### Author Response · Authors · 2023-11-23
> **Response to Reviewer 8D82**
>
> Thanks for your time reviewing and providing professional feedback for our manuscript. Our codes and models will be released, and here are responses to specific questions.
>
> **1. Lack of ablation experiments to validate the proposed model**
>
> Thanks for your question. We would like to clarify that we have conducted the suggested study in Sec. 4.2 and 4.3. Concretely,
> as explained in the last point of Sec. 4.2.2, we present 'LR' as a baseline, where we serialize pixels of the surface map as vectors of fMRI for brain decoding. This baseline is as a variant of our model where it excludes the fMRI pre-trained encoder. Therefore, results in Tabs. 2-4 are clear and enough to show the effectiveness of our pre-training on UBK dataset. We will make it clearer in the revision.
>
> **2. The training process is not adequately described**
>
> Thanks. We totally understand this concern. We have elaborated implement details in Sec. 4.1, including optimizer parameters and two-stage training way, to make our work reproducible. Moreover, our codes and models will be released upon acceptance.
>
> **3. Abbreviations are not clearly defined**
>
> We sincerely apologize for any confusion and typo. We will revise them and do careful proofreading. For your specific questions, (1) 'GLM' means general linear model; (2) there is a typo, words of 'batch size' should be deleted; (3) $E_{2}$ is the encoder in the second stage, where it takes as inputs quantized feature indices (stated in the first paragraph of Sec. 3.2.2). The input dimension is  $\in \mathbb{R}^{L}$, where $L$ is the length of indices, it is the same as the number of feature pixels output by the first stage encoder; (4) 'M', 'T' and 'V' in Tab. 4 indicate modality, text and vision, respectively.

---

### Meta-Review · Area_Chair_BTJW · 2023-12-08

**Metareview:**

The paper presents fMRI-PTE, an innovative autoencoder-based pretraining framework for fMRI data in brain decoding. Addressing the challenge of varying data dimensions across individuals, fMRI-PTE transforms fMRI signals into standardized 2D representations, ensuring dimensional consistency and preserving unique brain patterns. The self-supervised pre-training strategy enhances reconstruction quality for within-subject and cross-subject decoding tasks. Inspired by the VQGAN approach, fMRI-PTE demonstrates superior performance on the NSD dataset, marking a significant advancement in brain activity analysis with broad applications in neuroscience research.

The paper's weaknesses can be grouped into four key areas. Firstly, it faces criticism for insufficient empirical evaluation due to the absence of essential ablation experiments, unclear training process descriptions, and a presentation hindered by undefined abbreviations and lack of source code. Secondly, the paper is faulted for marginal methodological innovation, questioning the novelty of the proposed method, particularly the "innovative data transformation," and lacking clarity on the contributions of the training strategy and model architecture. The third set of weaknesses pertains to the weak support for foundational model and universal applicability, emphasizing the need for a robust baseline comparison, exploration of scaling and zero-shot capabilities, and support for claims about data transformation novelty. Lastly, there are concerns about the paper's lack of specificity and clarity, highlighting unclear research goals, methodological details, and the necessity for quantitative analyses like ablation studies. Addressing these issues is crucial for fortifying the paper and ensuring its suitability for publication.

**Justification For Why Not Higher Score:**

The paper's weaknesses can be grouped into four key areas. Firstly, it faces criticism for insufficient empirical evaluation due to the absence of essential ablation experiments, unclear training process descriptions, and a presentation hindered by undefined abbreviations and lack of source code. Secondly, the paper is faulted for marginal methodological innovation, questioning the novelty of the proposed method, particularly the "innovative data transformation," and lacking clarity on the contributions of the training strategy and model architecture. The third set of weaknesses pertains to the weak support for foundational model and universal applicability, emphasizing the need for a robust baseline comparison, exploration of scaling and zero-shot capabilities, and support for claims about data transformation novelty. Lastly, there are concerns about the paper's lack of specificity and clarity, highlighting unclear research goals, methodological details, and the necessity for quantitative analyses like ablation studies. Addressing these issues is crucial for fortifying the paper and ensuring its suitability for publication.

**Justification For Why Not Lower Score:**

N/A

---

### Decision · Program_Chairs · 2024-01-16

Reject